# SMART4Pain: Feasibility of a Two-Arm Pilot Study of an Integrated Rehabilitation Program for Adolescents and Their Parents for Improving Pain Management

Alessandro Failo [1,2,*], Paola Venuti [1], Patrizia Villotti [3], Stefan Schmalholz [2,4], Nicola Chistè [1], Bernd Raffeiner [5], Michela Salandin [6], Serena Pellegrin [6], Lydia Pescollderungg [7] and Mariantonietta Mazzoldi [2]

[1] Department of Psychology and Cognitive Sciences, University of Trento, Via Corso Bettini, 84, 38068 Rovereto, Italy; paola.venuti@unitn.it (P.V.); nicola.chiste@unitn.it (N.C.)
[2] Clinical Psychological Unit, Hospital of Bolzano, 39100 Bolzano, Italy; s.schmalholz@web.de (S.S.); mariantonietta.mazzoldi@sabes.it (M.M.)
[3] Career Counselling—Department of Education and Pedagogy, Université du Québec à Montréal, Montreal, QC H2X 3R9, Canada; villotti.patrizia@uqam.ca
[4] Family Counselling Centre, 39042 Bressanone, Italy
[5] Department of Rheumatology, Hospital of Bolzano, 39100 Bolzano, Italy; bernd.raffeiner@sabes.it
[6] Child Neurology Unit, Department of Pediatrics, Hospital of Bolzano, 39100 Bolzano, Italy; michela.salandin@sabes.it (M.S.); serena.pellegrin@sabes.it (S.P.)
[7] Department of Pediatrics, Hospital of Bolzano, 39100 Bolzano, Italy; lpescollderungg@hotmail.com
* Correspondence: a.failo@unitn.it

**Abstract:** Chronic pain in youth has an unsung etiology and limited treatment options. Affected adolescents show difficulties in different functioning domains, and their parents can develop associated distress, which negatively influences the adolescent's capacity to adjust to pain. The aims of this study are the following: (1) to develop an internet-delivered (online) pain intervention (SMART4Pain) program for adolescents and their parents, and to test its feasibility and acceptability; (2) to evaluate, in adolescents, the impact of the face-to-face, randomized, two-armed (i.e., CBT or biofeedback), open-label pilot study, developed together with the online program. The overall program consisted of six sessions scheduled over six weeks. Twenty adolescents (N = 20) and their parents (N = 20) completed the entire program and are included in this study. The results showed that all interventions were feasible and acceptable, as well as potentially effective in improving quality of life. Only the group receiving the biofeedback intervention showed some improvements in psychological indicators of stress. In conclusion, more research is needed to better understand and develop new, multimodal rehabilitation programs in outpatient settings.

**Keywords:** chronic pain; adolescents and parents; integrated treatment; online and face-to-face program; CBT; biofeedback



## 1. Introduction

Even though the exact extent of recurrent and chronic pain epidemiology in youth remains uncertain, one out of four minors are estimated to experience persistent or recurrent pain, or pain lasting longer than three months [1–3]. More evident in the scientific literature is that pain causes impaired functioning across a variety of domains, including physical, psychological, social, and developmental functioning, with severe consequences in everyday life and activities [1,4,5]. Pain is associated with an increased risk of anxiety [6–8], depression [9–11], school absences [12,13], social isolation, feelings of being misunderstood by healthy peers [14–16], sleeping issues with consequent fatigue [17], and overall poor quality of life [18–20]. Furthermore, chronic pain in youth is associated with a lifelong history of internalized mental health disorders reported in adulthood [21]. In this context, it is important to recall, from a long-term perspective, that medications used to alleviate pain and stress might prevent one from learning healthier ways of managing pain [22,23].

Unfortunately, only those suffering from severe levels of pain with a significant impact on daily functioning (3–5%) receive intense or sub-intensive rehabilitation treatments [24]. In terms of pain treatment in youth, the best recommendations are to privilege multi- and inter-disciplinary rehabilitation approaches [25]. More particularly, the world largest evidence base for chronic pain management in children recommends the use of psychological therapies, followed by pharmacological and physical therapies [26]; however, surprisingly, in reality, patients in pain usually have access to mainly physical therapy and medication treatments, and little–no access to psychological services, which is an aspect that multidisciplinary approaches should include [27].

Two recent Cochrane reviews are helpful for better understanding the quality of the results of psychoeducational treatments targeting youth. In their review, Fisher and colleagues [28] found that psychological treatments, delivered face-to-face, among children and adolescents (such as relaxation, hypnosis, coping skills training, biofeedback, and cognitive–behavioral therapy) are effective in reducing the intensity of chronic headache pain, recurrent abdominal pain, fibromyalgia, and sickle cell disease. In another review, Fisher and colleagues [29] found positive and satisfactory results for therapies delivered remotely (e.g., internet- and computer-based programs or smartphone applications). Despite encouraging results, the authors call for caution to be taken in interpreting these results, as the quality of the evidence is low or very low.

Pain conditions not only severely affect the life of the concerned individual, but also have spillover effects on the life of their close family members (parents, brothers, or sisters) in terms of decreased family functioning, increased stress and self-blame, and overall decreased quality of life [30,31]—compared with those families without children suffering from chronic pain [32–34]. This can, in turn, increase the risk of developing maladaptive behavioral responses to pain, and can play a role on the child's pain-related disability and adjustment [35]. Lastly, it seems to have an impact on the child's perception of pain [36].

Thus, as pointed out in the recent meta-analysis performed by Donnelly and colleagues [37], it seems important to promote changes in parenting behaviors in order to improve functional parent–child relationships and to help the child deal with chronic pain better.

Psychological therapies that can help with improving parenting behavior and reduce parent distress include problem-solving therapy (PST), cognitive–behavioral therapy (CBT), and problem-solving skills training (PSST) [38–41].

Some of these types of intervention (targeting the adolescent alone or both the adolescent and their parents) are offered face-to-face and are well implemented in the United States [42], Australia [43], the UK [44], and Germany [45].

Based on these considerations, we designed an intervention that simultaneously engages parents and adolescents, and that is offered both online (with the aim of providing flexibility, standardized information, and prompt communication with the patient and family) and face-to-face. Thus far, such an intervention has never been designed or offered in Italy. The specific aims of the present study are as follows: (1) to develop and test the feasibility and the acceptability of the SMART4Pain program; (2) to evaluate the impact of the face-to-face, randomized, two-armed, open-label pilot study, developed together with the online program.

The SMART4Pain program is an internet-delivered pain intervention, primarily based on rational–emotive education (REE)—a type of CBT designed from an educational perspective—for adolescents, and primarily based on problem-solving skills training (PSST) for their parents. These two transversal forms of skills for adolescents and parents, delivered in an online mode, are intended to provide them with a valuable background of knowledge and skills that they can draw on at any time of day, unencumbered by commitments; moreover, above all, it allows an optimization of time, without travel to the hospital.

Comparing two well-known in-person treatments (CBT and biofeedback) will help to better understand how they differ, with respect of several outcomes. For example, while

knowledge transfer from educational sources is probably linear within the psychotherapy context, the same may not be so true in the psychophysiological aspects related to stress. Moreover, the comparison of the two interventions may also better help to identify those underlying predisposing elements that could more effectively indicate which treatment works better for whom.

We hypothesized that both adolescents and parents participating in the study would report clinical improvements.

## 2. Experimental Section

### 2.1. Participants

The study sample includes 20 chronic pain parent–adolescent dyads. Participants were recruited during a 2 year period via their physician or psychologist, enrolled at the General Pediatric Unit, the Child Neurology Unit, the Department of Rheumatology, and the Clinical Psychological Unit in San Maurizio Regional Hospital of Bolzano (Italy).

Ethical approval was obtained from the Ethics Committee of Regional Hospital of Bolzano (n° 94-2017), by the University of Trento Human Subjects Research Committee (n° 2017-031). The study was conducted in accordance with ethical standards as laid down in the Declaration of Helsinki and its later amendments.

Inclusion criteria for participating in the study were as follows: (1) being an adolescent aged 11–18 years suffering from idiopathic chronic pain (defined as pain present for at least 3 months); (2) having at least one living parent (mother or father) willing to participate in the study; (3) having access to a web-enabled device (e.g., laptop, computer, tablet, or smartphone). Exclusion criteria were as follows: (1) having significant impairments in speaking or comprehension, or both; (2) having a cognitive impairment or an intellectual disability; (3) having an organic disease (such as cancer, diabetes, or cardiac diagnosis, etc.).

### 2.2. Study Flow

A total of 27 adolescents and their parents were contacted for eligibility, and 7 potential participant dyads were excluded, based on inclusion criteria. More specifically, 1 did not meet inclusion criteria due to presence of a chronic health condition; 3 families refused to participate due to lack of time or interest; and 3 families were unable to complete the treatment (for the reason of lacking time, either from the perspective of the adolescent or the parent, or both).

The final sample that completed the entire program, with all the primary outcome measures, consisted of 20 participants, randomly assigned to the cognitive–behavioral therapy (CBT) treatment (n = 10) or the biofeedback (BF) treatment (n = 10). Both treatment groups continued to receive their usual medical care during the whole duration of the program.

### 2.3. Procedures

We conducted an open-label, two-armed pilot study to test the feasibility and acceptability of a 6 week, non-intensive rehabilitation program called SMART4Pain, dedicated to adolescents with chronic pain and their parents. Assessments were conducted at pre-intervention and post-intervention (see Figure 1). Given both the researchers and participants knew which treatment would be administered, the two intervention groups (i.e., CBT and BF) were randomized before the enrollment of participants (with a 1:1 ratio), via random allocation software version 1.0 [46].

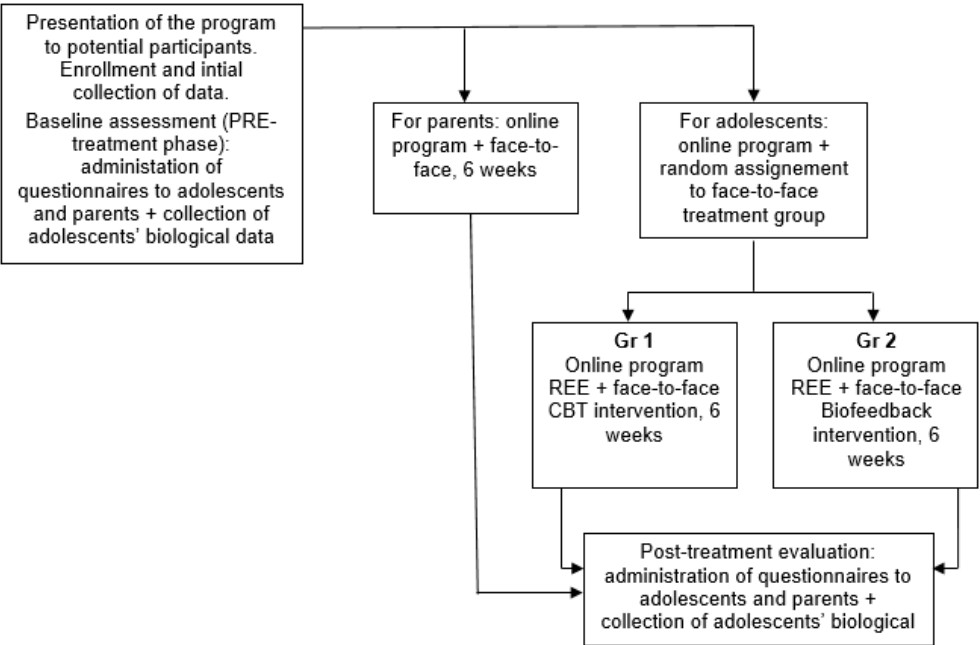

**Figure 1.** Graphical representation of the study's experimental design.

The program was divided into an online part (equal for everyone), and a face-to-face intervention part (different for each group). The online part for adolescents consisted of different content, primarily based on rational–emotive education (REE)—a positive, preventive, and interventionist psychological educational program, teaching rational critical thinking skills and effective problem-solving methods [47]. REE is based on both Bandura's social-learning theory (self-efficacy) and Beck's cognitive therapy (belief restructuring) [47–49]. The online part for the parents had different content, based on problem-solving skills training (PSST) [50,51]. Both these online materials were adapted specifically to chronic pain conditions. Figures 2 and 3 summarize the online content of the SMART4Pain program for adolescents and their parents. The face-to-face part was delivered to the parents, targeting their needs, and to the 2 groups of treatment for adolescents as follows: A—different CBT skills applied to their needs; B—biofeedback treatment. The structure of the whole SMART4Pain program is shown in Figure 4.

| Adolescents | Content | Quiz at the end | Activity at the end of the module |
|---|---|---|---|
| 1 – Informations about pain | Understanding chronic pain: how the body processes the pain signals, pain can be modified, how to fix objectives to handle pain, the importance of positive attitude when enrolling in the program. | Yes | Definition of objectives<br><br>Pain diary (weekly) |
| 2 – What can be initially done to handle pain | The role played by thoughts, how to identify positive and negative thoughts, how to reduce negative thoughts, the concept of neuroplasticity in modifying pain components. | Yes | Pain triggering events<br><br>Pain diary (weekly) |
| 3 – Role of stress and emotions | What stress is and how to handle it, the role played by negative emotions in making pain worse, how to identify and handle emotions to better handle pain. | Yes | Emotions and pain<br><br>Pain diary (weekly) |
| 4 – Relaxation strategies and sleep cycles management | Relaxation strategies (breating, imagination, muscular relaxation), distraction. How to improve quality of sleep, as it influence chronic pain. | Yes | Relaxation and sleep<br><br>Pain diary (weekly) |
| 5 – School, life style and activities | How pain interferes with school and how to handle it, how to obtain parents', teachers' and friends' support. The importance of life style and to do different activities regularly. | Yes | Activities planification<br><br>Pain diary (weekly) |
| 6 – Competences maintenance and relapse prevention | Recap of attained ojectives and identification of areas that need to be improved, the tools to handle pain, the identification of barriers and obstacles. Prevention strategies. Synthesis of all learned techniques. | No | Final questionnaire of the program<br><br>Pain diary (weekly) |

**Figure 2.** Description of the online content (for adolescents) of the SMART4Pain program.

| Parents | Content | Activity at the end of the module |
|---|---|---|
| 1 – Informations about pain | How the program is structured, information about chronic pain, how to set up the objectives, the importance of a positive attitude in approaching the program to help the adolescent. | Definition of objectives |
| 2 – How to implement problem solving strategies | The role of thoughts, the processes and phases of problem solving (optimism, identification of the problem, definition of possible options, analysis of options, choosing an option, evaluation of the adopted strategy). | Problem Solving (basic) |
| 3 – Stress and emotions in problem solving processes | The role played by stress and emotions in worsening pain, the role of emotions and thoughts in problem solving process, how stress and emotion affects family members and the pain of the adolescent, how to regulate stress. | Problem Solving (advanced) |
| 4 – Relaxation strategies and sleep cycles management | The importance of relaxation and of adaptive coping strategies. The link between pain and sleep, how to help the adolescent to ameliorate his/her sleeping. | Relaxation and sleep |
| 5 – How to motivate and communication strategies | Prepare a plan for school by designing a system of expectations and reinforcement to empower. Lifestyles and chronic pain. Communication strategies to foster independence. | Expectations and reinforcement plan |
| 6 – Competences maintenance and relapse prevention | Areas of success and remaining challenges, prepare plans for the future, ten tips to keep in mind. Final summary of all activities at a glance. Summary of goals achieved and areas still needing improvement, tools to manage pain, identification of barriers and obstacles. Prevention and how to prepare a plan for the future. Summary of all techniques learned. | Final questionnaire of the program |

**Figure 3.** Description of the online content (for parents) of the SMART4Pain program.

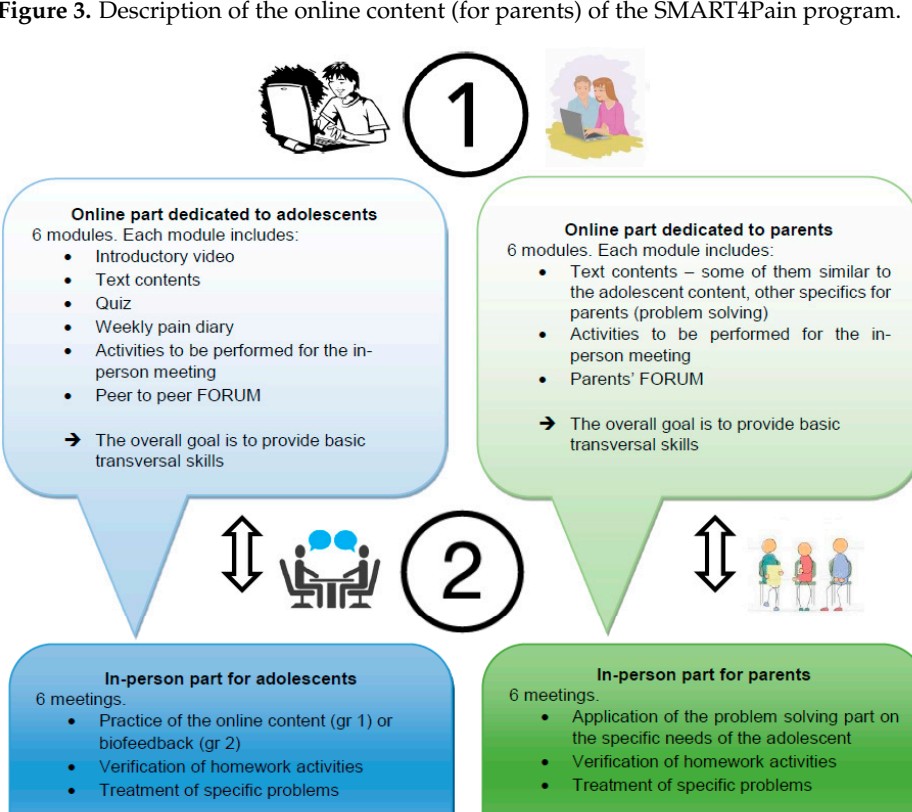

**Figure 4.** The structure of the SMART4Pain program.

To implement the online part of the program, six modules, based on a psychoeducational perspective, were used. These modules provided weekly, home-based parent–adolescent activities for both the parents and the adolescents, focusing on behavior changes, problem solving approaches, enhanced functional abilities, and further training and resources for coping with pain, as well as improving social support. Alongside the online modules, the adolescents and the parents had free access to dedicated chat rooms or forums, which were moderated by the program therapists.

Concerning the face-to-face intervention part of the program, adolescents were informed which group they would be in (i.e., biofeedback or CBT) after completing the pre-treatment assessment. The difference between the two intervention groups was the type of therapy the adolescent received.

In group A, the cognitive–behavioral therapy (CBT) skills and tools were adapted to the needs and problems experienced by the adolescent, by goal setting, thought restructuring, diaphragmatic breathing, guided imagery, planning motivation activities, and social or peer support.

In group B, the central point was to help participants deal with the stress, using a biofeedback device with software and different sensors that teach adolescents how to have better body control and how to decrease anxiety (e.g., how to monitor the heart rate rhythm or breathing patterns (or both) and to modulate muscle tension, etc.).

All parents received a tailored intervention face-to-face, specifically targeting their own adolescent's characteristics, primarily based on a version of problem-solving skills training (PSST), adapted for chronic pain conditions [40,41]. This approach is based on the social problem-solving model, with specific activity pacing for returning to function [50], as well as teaching a structured approach to solving problems.

The face-to-face phase for the adolescent and one of his or her parents consisted of 6 sessions (1 h, once a week for each of them), scheduled over 6 weeks. In few cases (n = 5), the participants postponed one or more appointment, making the duration of the intervention increase to 8 weeks. During the same session, one therapist worked with the adolescent and another worked with parent to improve time scheduling.

### 2.4. Outcome Measures

#### 2.4.1. Feasibility

The feasibility of the program was assessed using four metrics, as follows: (1) the study recruitment or enrollment statistics; (2) the treatment adherence, as assessed by the number of online modules that were completed by adolescents and their parents and the number of rescheduled face-to-face treatment sessions; (3) the percentage of use of online modules instead of other modalities (e.g., a request to send material by email or printing it); (4) the use of the online chat forum function.

#### 2.4.2. Treatment Acceptability and Satisfaction

At the end of the program, parents completed a version of the nine-item Treatment Evaluation Inventory—Short Form (TEI-SF), adapted for chronic pain conditions [51], to assess acceptability and satisfaction, which focused on parents' experience with the entire program [52]. Response options range from 1 (strongly disagree) to 5 (strongly agree). These were summed to create a total score ranging from 9 to 45—a "moderate" satisfaction and acceptability of a given treatment was indicated by a score of 27 or higher [52,53]. Furthermore, parents were asked to evaluate the improvement of the prefixed activities using a 6-point rating scale (0—not at all likely; 6—extremely likely).

Adolescents completed a measure of treatment expectancy and treatment satisfaction by responding to a short questionnaire, where they rated the percentage of attained and improved goals, the level of satisfaction, and the perceived utility of the program. The satisfaction and utility scales were rated on a 10-point Likert scale (0—did not like, 10—extremely liked). In addition, adolescents had the chance to respond to an open question

in order to better explain their point of view (i.e., they were invited to tell, in more detail, what they liked and what they disliked).

2.4.3. Pre-Post Treatment Measures
Demographics and Pain Characteristics

As part of the initial semi-structured interview, we collected basic demographic information from parents (e.g., age, gender, and type of primary diagnosis). From adolescents, we collected data on their (a) pain severity (0—no pain; 10—most pain possible); (b) pain frequency (days per week, 0–7); (c) pain duration (3–6 months; > 6 months); (d) pain location (1 site of pain; 2 or more sites of pain).

Parenting Role Stress

Parents completed the 36-item Parenting Stress Index—Short Form [54]. Responses ranged from 0 (strongly agree) to 5 (strongly disagree). Items were combined to create a total stress percentile score, with higher scores reflecting greater parenting role stress. The PSI-SF produces a total stress score, which has a clinical cutoff of 90, as well as 3 subscales (plus 1 for control, for a total of 4 subscales) as follows: Parental Distress, Parent–Child Dysfunctional Interaction, Difficult Child, and Defensive Response. The Defensive Response is a control subscale, which indicates whether the parent is presenting a "minimizing" or "look good" bias to their responses. This tool has been largely used with parents of adolescents with chronic pain, e.g., as discussed in Refs. [40,55].

Pain Diary

A daily diary was used to assess self-reported pain information (presence and intensity of pain, duration of pain, and use of analgesic on demand) during the program period. Pain intensity was assessed using an 11-point numerical rating scale (NRS), with anchors from 0 (no pain) to 10 (worst pain). Pain duration was evaluated by the number of min/h of daily pain. The use of analgesic on demand was identified by the number of medications taken per day.

QUID

The Italian Pain Questionnaire (QUID) [56], is a self-report instrument assessing quality of pain perception, and it is a reconstructed Italian version of the McGill Pain Questionnaire [57]. It represents the most parsimonious, meaningful, and idiomatic set of Italian pain descriptors, providing quantitative information that can be treated statistically. It includes a semantic interval scale consisting of 42 pain descriptors and is divided into four main classes as follows: sensory, affective, evaluative, and mixed. The items are combined to provide a total pain score. It can also be used in developmental age [58].

PedsQL 4.0

The Pediatric Quality of Life Inventory (PedsQL 4.0) [59] is a multidimensional child or adolescent self-report scale (and also parent proxy report used in this study, only to evaluate the congruence with perceptions of their daughter or son) for measuring health-related quality of life (HRQOL). It consists of 23 questions, which cover four domains, as follows: physical, emotional, social, and school functioning, by a 5-point response scale. A domain-specific score is calculated from the corresponding questions, ranging from 0 (worst HRQOL) to 100 (best HRQOL), which can be combined for a total functioning score. Higher scores indicate fewer difficulties—a better health-related quality of life.

TAD

The Adolescent and Childhood Anxiety and Depression Test (TAD) is a questionnaire measuring anxiety and depression in developmental age [60]. The TAD comprises three rating scales, as follows: student, teacher, and parent. For the purpose of this study, we administered only the student and parent rating scales, but in the analysis, we considered

only the self-report scale. The TAD is a 22-item questionnaire, which requires responses on a 5-point Likert scale. To compute the scoring, 11 items contribute to the definition of an index of depression, while the remaining 11 furnish an index of anxiety. Both indexes have a mean of 100.

CRI—Coping Responses Inventory

The Coping Response Inventory—Youth Form (CRI-Y) [61] is a 48-item, brief self-report inventory, that measures and assesses the cognitive and behavioral responses that adolescents use to cope with a recent problem or stressful situation, such as pain. The CRI-Y has eight scales covering the areas of Approach Coping Styles (Logical Analysis, Positive Reappraisal, Seeking Guidance and Support, and Problem Solving) and Avoidant Coping Styles (Cognitive Avoidance, Acceptance or Resignation, Seeking Alternative Rewards, and Emotional Discharge). The adolescent is asked to identify his or her responses to a previously identified pain situation indicating how often he or she took the described action to deal with the problem. Responses are on a 4-point rating scale, ranging from never (0) to fairly often (3).

2.4.4. Physiological Measures

For participants receiving the biofeedback face-to-face intervention, physiological data were collected, as follows.

Electrodermal Activity (SCL-GSR)

Skin conductance measurements reflect changes in sweat gland activity, caused by the activity in the sympathetic nervous system of postganglionic cholinergic fibers. This is usually associated with cortical activation of anxious thoughts, worries, and stress. It has been one of the most used parameters in the past, able to reflect emotional changes following conditions of discomfort, such as those caused by anxiety or pain, e.g., as discussed in Refs. [62–64].

Muscular Tension (sEMG)

Surface electromyography records the bioelectric activity produced by the muscle when it contracts. The electrical signal, recorded on the surface of the skin (sEMG), is the result of the simultaneous activity of numerous motor units that are under voluntary controlled and, thus, the EMG signal is easily controlled voluntarily. It is a parameter typically used in musculoskeletal pain, e.g., as discussed in Refs. [65–67].

Heart Rate Variability (HRV SDNN)

Cardiac variability is an index of a wellbeing characteristic of a physiological system that adapts easily to stressful situations. The two parameters considered in cardiac variability are amplitude (representing quantity) and coherence (representing quality). In this study, we chose to use (in the time domain) the SDNN (msec) parameter—that is, the standard deviation (SD) of the "normal to normal" (NN) intervals. The more SDNN is reduced, the greater the sympathetic prevalence will be. HRV biofeedback allows for the management of imbalances in the autonomic nervous system (ANS) by regulating afferent vagal tones that inhibit the flow of pain signals traveling to the brain, e.g., as discussed in Refs. [68–70].

Temperature (TEMP)

Measurement of skin temperature in the peripheral area reflects the blood flow in the vessels under the skin. The change of this parameter is associated with sympathetic activity: it is an indicator of physiological flexibility and health. The change of this indicator contributes to relaxation and stress management by improving blood circulation. It is a useful signal for certain types of headaches and circulation-related disorders, e.g., as discussed in Refs. [71,72].

Respiratory Sinus Arrhythmia (RSA)

Respiratory sinus arrhythmia represents an indicator of how much the heart rate fluctuates during breathing. A low RSA identifies a lower adaptation to physical performance but also reflects on mental performance, hindering the search for the optimal strategy to cope with a given situation. Faced with a pain that has peaks of onset, we tend to hyperventilate; therefore, this parameter is useful to understand how to learn a voluntary control of breathing patterns, e.g., as discussed in Refs. [73–75].

## 3. Results

### 3.1. Demographic and Pain Characteristics at Baseline

Sample characteristics are shown in Table 1. Participants included 20 adolescents aged 11–18 years (M = 14.73, SD = 1.97) and 20 parents aged 39–54 years (M = 46.97, SD = 4.71). Adolescent participants were primarily female (85%) and were referred to the study for treatment of head pain (55%), musculoskeletal pain (35%), and abdominal pain (10%). The day they started the program, they showed a moderate level of pain (M = 5.85, SD = 1.69). Pain was present, on average, 4 days in a week (M = 4.30, SD = 2.66). The onset of pain was, for the most part, more than 6 months ago (n = 14.70%) and in most cases pain involved two or more sites (n = 13.65%). Parent participants were primarily mothers (90%).

### 3.2. Feasibility of the Program, Treatment Acceptability, and Satisfaction

Descriptive information about program usage and feasibility is shown in Table 2. Retention rate (participants retained and assessed with valid outcome data) in the study was 74% (20 family on 27 contacted) and all participants completed the 6 modules of the program in a timeframe of 6–8 weeks. More specifically, 5 participants (20%) needed to postpone the clinical appointment one or two times to a subsequent date. About half of the sample (55%) made use of the online modules instead of other modalities. A minority (25%) made use of the chat forum. Participants showed an overall satisfaction and utility of the program, and obtained the primary objectives set. Parents showed a more than moderate satisfaction and acceptability of the program with a mean score of 32.55 reported at the TEI-SF and an improvement of the prefixed activities (M = 4.1, SD = 0.91).

### 3.3. Pre–Post Treatment Measures

Pain Diary (Intensity, Duration, and Medication Usage)

The Friedman's ANOVA [76] was used to test for differences in time with respect to pain intensity, pain duration, and use of medication in the sample of adolescents. Pain intensity ($X^2_F(5) = 1.59$, $p = 0.903$, Table 3), pain duration ($X^2_F(5) = 2.15$, $p = 0.828$, Table 4), and use of medications ($X^2_F(5) = 6.07$, $p = 0.300$, Table 5) of participants did not significantly change over time.

Parenting Role Stress

The Wilcoxon's rank sum test was applied to evaluate changes in parental stress before and after treatment. As shown in Table 6, no significant differences were detected.

QUID

The Wilcoxon's rank sum test was applied to evaluate changes in pain intensity, comparing baseline and follow-up levels. As shown in Table 7, no significant differences were detected.

PedsQL 4.0

Quality of life scores of adolescents were compared before and after treatment. On average, a better overall quality of life was obtained after treatment (Mdn = 72.25) compared to pre-treatment score (Mdn = 67.25). A Wilcoxon signed rank test indicated that this difference was statistically significant, T = 43.00, z = −2.32, $p < 0.05$. Some improvement was seen in school functioning domain scores ($p < 0.05$). See Table 8 for details.

**Table 1.** Sociodemographic and clinical characteristics of participants at baseline.

| Baseline Characteristic | M [SD] or n (%) | HEAD * (n = 11) M [SD] or n (%) | MSK * (n = 7) M [SD] or n (%) | RAP * (n = 2) M [SD] or n (%) | CBT * (n = 10) M [SD] or n (%) | BF * (n = 10) M [SD] or n (%) |
|---|---|---|---|---|---|---|
| Adolescents sample (n = 20) | | | | | | |
| Age | 14.73 [1.97] | 14.24 [2.17] | 15.24 [1.87] | 15.6 [0.28] | 15.17 [2.03] | 14.28 [1.91] |
| Gender | | | | | | |
| Female | 17 (85) | 9 (81.8) | 7 (100) | 1 (50) | 10 (100) | 7 (70) |
| Male | 3 (15) | 2 (18.2) | 0 (0) | 1 (50) | 0 (0) | 3 (30) |
| Primary diagnosis | | | | | | |
| Head pain | 11 (55) | 11 (100) | 0 (0) | 0 (0) | 5 (50) | 6 (60) |
| Musculoskeletal pain | 7 (35) | 0 (0) | 7 (100) | 0 (0) | 4 (40) | 3 (30) |
| Abdominal pain | 3 (10) | 0 (0) | 0 (0) | 2 (100) | 1 (10) | 1 (10) |
| Group allocation | | | | | | |
| CTB | 10 (50) | 5 (45.5) | 4 (57.1) | 1 (50) | 10 (100) | 0 (0) |
| biofeedback | 10 (50) | 6 (54.5) | 3 (42.9) | 1 (50) | 0 (0) | 10 (100) |
| Intensity of pain at baseline | 5.85 [1.69] | 5.63 [1.69] | 6.00 [2.00] | 6.50 [0.71] | 5.40 [1.58] | 6.30 [1.77] |
| Weekly frequency of pain | 4.30 [2.66] | 3.55 [2.66] | 5.29 [2.63] | 5.00 [2.83] | 3.60 [2.63] | 5.00 [2.62] |
| First pain episode | | | | | | |
| 3–6 months | 6 (30) | 4 (36.4) | 0 (0) | 2 (100) | 3 (30) | 3 (30) |
| >6 months | 14 (70) | 7 (63.6) | 7 (100) | 0 (100) | 7 (70) | 7 (70) |
| Nr. of pain sites | | | | | | |
| One | 7 (35) | 6 (54.5) | 0 (0) | 1 (50) | 3 (30) | 4 (40) |
| Two or more | 13 (65) | 5 (45.4) | 7 (100) | 1 (50) | 7 (70) | 6 (60) |
| Parents sample (n = 20) | | | | | | |
| Age | 46.97 [4.71] | | | | | |
| Gender | | | | | | |
| Female | 18 (90) | | | | | |
| Male | 1 (5) | | | | | |
| Female, male | 1 (5) | | | | | |

* RAP—recurrent abdominal pain; MSK—musculoskeletal pain; HEAD—primary headache syndrome (migraine or tension-type headache); CBT—cognitive–behavioral therapy; BF—biofeedback.

**Table 2.** Program feasibility and treatment adherence.

| Program Feasibility and Treatment Adherence | | HEAD (n = 11) | MSK (n = 7) | RAP (n = 2) | CBT (n = 10) | BF (n = 10) |
|---|---|---|---|---|---|---|
| | M [SD] or n (%) | M [SD] or n (%) | M [SD] or n (%) | M [SD] or n (%) | M [SD] or n (%) | M [SD] or n (%) |
| Adolescents sample (n = 20) | | | | | | |
| Treatment adherence | | | | | | |
| N of online modules completed | 6 (100) | 6 (100) | 6 (100) | 6 (100) | 6 (100) | 6 (100) |
| Use of online modules | | | | | | |
| Yes | 11 (55) | 7 (63.6) | 3 (42.9) | 1 (50) | 7 (70) | 4 (40) |
| Yes, partially | 1 (5) | 1 (9.1) | 0 (0) | 0 (0) | 0 (0) | 1 (10) |
| No | 8 (40) | 3 (27.3) | 4 (57.1) | 1 (50) | 3 (30) | 5 (50) |
| Use of the chat | | | | | | |
| Yes | 5 (25) | 3 (27.7) | 2 (28.6) | 0 (0) | 3 (30) | 2 (20) |
| No | 15 (75) | 8 (72.7) | 5 (71.4) | 2 (100) | 7 (70) | 8 (80) |
| Primary objectives attained (0–100) | 73.75 [25.28] | 68.18 [20.41] | 75 [32.28] | 100 [0] | 77 [24.528] | 70.5 [26.92] |
| Overall satisfaction (0–10) | 6.90 [2.17] | 7 [1.73] | 6.29 [2.93] | 8.50 [0.71] | 6.60 [2.41] | 7.20 [1.99] |
| Overall utility (0–10) | 6.50 [3] | 6.55 [3.30] | 6 [3.06] | 8 [0.00] | 6 [3.30] | 7 [2.75] |
| Parents sample (n = 20) | | | | | | |
| Increased activity (1–6) | 4.1 [0.91] | | | | | |
| TEI-SF (cut-off > 27) | 32.55 [2.50] | | | | | |

**Table 3.** Pain intensity (N = 20).

| Pain Intensity (0–10) | Week 1 | | Week 2 | | Week 3 | | Week 4 | | Week 5 | | Week 6 | | *p*-Value |
|---|---|---|---|---|---|---|---|---|---|---|---|---|---|
| | Min–Max | M (SD) | Min–Max | M (SD) | Min–Max | M (SD) | Min–Max | M (SD) | Min–Max | M (SD) | Min–Max | M (SD) | |
| Total sample (n = 20) | 2.7–8.7 | 5.76 (1.88) | 2.0-8.1 | 5.81 (1.71) | 3.0–8.2 | 6.01 (1.64) | 0.0–8.4 | 5.56 (2.12) | 0.0–8.5 | 5.52 (2.13) | 0.0–8.7 | 5.64 (2.19) | 0.903 |
| CBT | 2.7–8.3 | 5.28 (1.90) | 3.7–7.6 | 5.62 (1.39) | 3.7–7.8 | 5.76 (1.19) | 4.0–7.6 | 5.50 (1.33) | 3.7–7.5 | 5.33 (1.34) | 3.9–7.5 | 5.03 (1.35) | 0.736 |
| BF | 2.8–8.7 | 6.23 (1.82) | 2.0-8.1 | 5.99 (2.04) | 3.0–8.2 | 6.25 (2.02) | 0.0–8.4 | 5.62 (2.77) | 0.0–8.5 | 5.71 (2.77) | 0.0–8.7 | 5.70 (2.83) | 0.729 |

**Table 4.** Pain duration (N = 20).

| Pain Duration (0–24) | Week 1 | | Week 2 | | Week 3 | | Week 4 | | Week 5 | | Week 6 | | *p*-Value |
|---|---|---|---|---|---|---|---|---|---|---|---|---|---|
| | Min–Max | M (SD) | Min–Max | M (SD) | Min–Max | M (SD) | Min–Max | M (SD) | Min–Max | M (SD) | Min–Max | M (SD) | |
| Total sample (n = 20) | 1–20 | 10.5 (5.91) | 1.5–20 | 10.5 (5.91) | 1–20 | 10.5 (5.91) | 1.5–20 | 10.5 (5.91) | 1–20 | 10.5 (5.91) | 1.5–20 | 10.5 (5.91) | 0.828 |
| CBT | 3–19 | 11.60 (5.82) | 1.5–19 | 10.55 (6.47) | 1–19 | 11.35 (6.45) | 1.5–19 | 10.90 (6.72) | 1–19 | 11.75 (6.04) | 1.5–20 | 11.95 (6.42) | 0.273 |
| BF | 1–20 | 9.4 (6.09) | 3.5–20 | 10.45 (5.65) | 2.5–20 | 9.65 (5.52) | 3–20 | 10.10 (5.32) | 2–20 | 9.25 (5.82) | 1.5–17.5 | 9.05 (5.28) | 0.341 |

**Table 5.** Use of medications (N = 20).

| Use of Medications | Week 1 | | Week 2 | | Week 3 | | Week 4 | | Week 5 | | Week 6 | | *p*-Value |
|---|---|---|---|---|---|---|---|---|---|---|---|---|---|
| | Min–Max | M (SD) | Min–Max | M (SD) | Min–Max | M (SD) | Min–Max | M (SD) | Min–Max | M (SD) | Min–Max | M (SD) | |
| Total sample (n = 20) | 4.5–20 | 10.5 (5.70) | 5–20 | 10.5 (5.56) | 5–20 | 10.5 (5.61) | 8–20 | 10.5 (4.50) | 7.5–20 | 10.5 (4.79) | 8.5–20 | 10.5 (4.14) | 0.300 |
| CBT | 4.5–18.5 | 8.55 (5.19) | 5–18.5 | 9.27 (5.21) | 5–17.5 | 8.91 (4.86) | 8–18 | 8.91 (3.02) | 7.5–17.5 | 10.05 (4.39) | 8.5–18 | 10.14 (3.65) | 0.192 |
| BF | 4.5–13 | 8.75 (6.01) | 5–12.5 | 8.75 (5.30) | 5–17.5 | 11.25 (8.84) | 8–8 | 8 (0.00) | 7.5–7.5 | 7.5 (0.00) | 8.5–8.5 | 8.5 (0.00) | 0.893 |

**Table 6.** Parenting Role Stress scores at baseline and post-treatment (N = 20).

| Variable | Pre-Treatment M (SD) | Post-Treatment M (SD) | Z | Wilcoxon Signed Rank Test Probability |
|---|---|---|---|---|
| Parental distress | 39.30 (25.77) | 36.05 (25.58) | −0.835 | 0.404 |
| Parent–child dysfunctional interaction | 52.35 (18.50) | 49.70 (21.18) | −0.869 | 0.385 |
| Defensive response | 48.75 (30.34) | 48.00 (29.63) | −0.131 | 0.896 |
| Difficult Child | 64.70 (23.21) | 61.00 (25.80) | −0.769 | 0.442 |
| Total Stress | 53.85 (24.13) | 49.75 (24.95) | −1.15 | 0.250 |

**Table 7.** QUID scores at baseline and post-treatment (N = 20).

| Variable | Pre-Treatment M (SD) | Post-Treatment M (SD) | Z | Wilcoxon Signed Rank Test Probability |
|---|---|---|---|---|
| QUID Total score | 29.70 (15.53) | 29.50 (11.25) | −0.175 | 0.861 |
| Sensory | 12.10 (5.73) | 11.50 (5.35) | −0.508 | 0.612 |
| Affective | 5.40 (2.87) | 5.90 (2.38) | −0.727 | 0.467 |
| Evaluative | 7.85 (5.31) | 8.85 (5.47) | −0.826 | 0.409 |
| Mixed | 4.40 (3.69) | 3.25 (2.19) | −1.346 | 0.178 |

**Table 8.** PedsQL 4.0 (N = 20).

| Variable | Pre-Treatment M (SD) | Post-Treatment M (SD) | Z | Wilcoxon Signed Rank Test Probability |
|---|---|---|---|---|
| Total Quality of life (0–100) | 63.63 (17.91) | 71.01 (17.15) | −2.32 | 0.021 |
| Physical (0–100) | 58.67 (23.46) | 66.37 (23.67) | −1.32 | 0.188 |
| Emotional (0–100) | 60.75 (25.09) | 69.0 (21.19) | −1.83 | 0.067 |
| Social (0–100) | 81.25 (24.32) | 84.25 (16.80) | −0.738 | 0.461 |
| School functioning (0–100) | 51.45 (24.54) | 65.5 (25.49) | −2.31 | 0.021 |

TAD

The Wilcoxon's rank sum test was applied to evaluate changes in anxiety and depression symptoms, comparing baseline and follow-up levels. As shown in Table 9, no significant differences were detected.

**Table 9.** Anxiety and Depression (N = 20).

| Variable | Pre-Treatment M (SD) | Post-Treatment M (SD) | Z | Wilcoxon Signed Rank Test Probability |
|---|---|---|---|---|
| Anxiety | 100 (18.21) | 97.50 (17.51) | −1.02 | 0.310 |
| Depression | 100.25 (17.36) | 97.00 (14.46) | −1.24 | 0.217 |

CRI—Coping Responses Inventory

The Wilcoxon's rank sum test was applied to evaluate changes in the coping responses comparing baseline and follow-up levels. As shown in Table 10, no significant differences were detected.

**Table 10.** Coping responses inventory (N = 20).

| Variable | Pre-Treatment M (SD) | Post-Treatment M (SD) | Z | Wilcoxon Signed Rank Test Probability |
|---|---|---|---|---|
| Approach coping styles | | | | |
| Logical analysis | 44.20 (11.61) | 44.05 (9.09) | −0.078 | 0.938 |
| Positive Reappraisal | 48.70 (13.96) | 51.10 (12.01) | −1.54 | 0.123 |
| Seeking Guidance and Support | 53.30 (11.55) | 52.70 (9.00) | −0.370 | 0.711 |
| Problem Solving | 49.65 (12.98) | 53.50 (12.26) | -1.54 | 0.123 |
| Avoidant coping styles | | | | |
| Cognitive Avoidance | 49.55 (10.93) | 49.05 (9.27) | −0.371 | 0.711 |
| Acceptance or Resignation | 50.25 (11.36) | 50.10 (10.35) | −0.189 | 0.850 |
| Seeking Alternative Rewards | 45.05 (9.92) | 47.50 (13.33) | −0.976 | 0.329 |
| Emotional Discharge | 45.85 (11.27) | 46.10 (8.25) | −0.142 | 0.887 |

Physiological Measures

Table 11 reports the results of the Wilcoxon signed rank test performed on pre- and post-treatment measures in adolescents. The heart variability and the respiratory sinus arrhythmia scores significantly improved over time.

**Table 11.** Physiological measures (N = 10).

| Variable | Pre-Treatment M (SD) | Post-Treatment M (SD) | Z | Wilcoxon Signed Rank Test Probability |
|---|---|---|---|---|
| Electrodermal activity (SCL-GRS) | 5.14 (3.14) | 3.75 (2.91) | −1.68 | 0.09 |
| Muscular tension (sEMG) | 5.37 (2.38) | 4.59 (2.68) | −0.866 | 0.386 |
| Heart Rate Variability (HRV SDNN) | 72.95 (18.86) | 100.80 (48.06) | −2.29 | 0.02 |
| Temperature | 31.09 (4.17) | 32.22 (2.74) | −0.866 | 0.386 |
| Respiratory Sinus Arrhythmia (RSA) | 17.60 (5.52) | 31.94 (19.03) | −2.80 | 0.005 |

## 4. Discussion

To our knowledge, this is the first study in Italy to examine an educational, psychological–psychophysiological intervention that targets wellbeing improvement in adolescents with chronic pain and their parents in a non-intensive pain rehabilitation setting. The present study described the preliminary findings concerning the participation in a mixed program (online and face-to-face), called SMART4Pain, in order to evaluate its feasibility and acceptability. We investigated also the potential efficacy of this program specifically tailored for the needs of this population.

The results showed that the intervention was feasible and acceptable, as well as potentially efficacious in improving quality of life, consistent with other studies [25,35,39,77]. The adolescents have reported an overall satisfaction and found the treatment to be useful. The majority of parents perceived the treatment as credible, reported a high degree of

satisfaction with the entire program, and showed a good improvement of the prefixed activities, which is consistent with other similar study [35,78,79]. They have also showed a good adhesion to scheduled treatment sessions.

Immediately post-program, the data showed that adolescents reported no significant differences in pain characteristics (intensity, duration, quality), in anxiety and depression traits, or coping strategies, and there was no significant changes in parenting distress, unlike other similar studies [38,44,51,80,81]. Surprisingly, no statistically significant changes pre-post treatment were found for the whole program, except in the improvement of adolescents' quality of life (overall and in the school functioning domain), despite the scientific literature showing the efficacy of self-management (internet) and cognitive-behavioral therapy intervention programs for adolescents with chronic pain and their parents. More specifically, Fisher and colleagues suggest that when both goals match at the beginning of the treatment, a lower intensity of pain can be expected post-treatment and at follow-up [82]. Thus, the focus on rational–emotive education of our pilot study may need to be revisited and improved. While they were in the expected direction of grasping differences between the proposed in-person treatments, the results yielded to no statistical significance. A larger sample size is warranted to draw further conclusions. However, the biofeedback intervention group showed some improvements in psychological indicators of stress, in agreement with some of the literature e.g., [10,71,72].

As a review suggested [83], it seems that biofeedback can be a promising intervention for stress management. In particular, this study is one of the few which has explored the implementation of biofeedback in outpatient contexts and in samples of adolescents with pain associated to various conditions. Furthermore, our protocol adds, to the literature, some reflections about the relationship between psychological outcomes (through our psychoeducational internet-delivered pain intervention) and physiological outcomes of biofeedback. Thus, this intervention could be relevant to adolescents' lives, since it has the potential to improve self-regulation skills in chronic pain management and to translate these skills to real-world settings [84].

Our study has several limitations that should be taken into account. A first important limitation is that a follow-up is missing. Unfortunately, the clinical context and some restrictions made it impossible to implement a longitudinal study design that would have allowed us to draw more solid conclusions. Furthermore, the size of our sample may represent a limitation of our statistical analyses. The brief nature of the proposed interventions (6–8 weeks in total) may also not be sufficiently long enough to establish lasting changes. Further replication is needed to verify the results of this study in a randomized controlled trial.

Despite these limitations, this study provides a better understanding of the strategies that are suitable for development in the context of mixed (online and in-person) interventions. This approach is lacking in the current literature: while internet and face-to-face interventions should be both available to all adolescents with a chronic pain condition, common practices are distant from reaching this potential [85].

The strengths of this study include the comprehensive assessment of feasibility and outcomes in several important domains. Moreover, the use of a standardized part of the treatment ensured that all participants took part in the same treatment content.

In conclusion, there is an urgent need for more research to help developing and evaluating new multimodal rehabilitation programs in outpatient settings. This is also necessary to improve the applicability of these treatments to usual care settings.

**Author Contributions:** Conceptualization, A.F., P.V. (Paola Venuti) and M.M.; methodology, A.F., P.V. (Paola Venuti), N.C. and M.M.; software, A.F. and P.V. (Paola Venuti); validation, A.F., P.V. (Paola Venuti) and M.M.; formal analysis, P.V. (Paola Venuti); investigation, A.F. and S.S.; resources, A.F., P.V. (Paola Venuti) and N.C.; data curation, A.F., P.V., M.M. and P.V. (Patrizia Villotti); writing—original draft preparation, A.F. and P.V. (Patrizia Villotti); writing—review and editing, A.F. and P.V. (Patrizia Villotti); visualization, A.F., P.V. (Paola Venuti), P.V. (Patrizia Villotti), S.S., N.C., B.R., M.S., S.P., L.P., M.M.; supervision, A.F., P.V. (Paola Venuti) and M.M.; project administration, A.F., P.V. (Paola Venuti)

and M.M.; funding acquisition, P.V. (Paola Venuti). All authors have read and agreed to the published version of the manuscript.

**Funding:** At the time that the research was being implemented, it was funded by Fondazione Trentina per la Ricerca sui Tumori and University of Trento (no grant number). Alessandro Failo is actually supported by the Rheuma-Liga Südtirol for the clinical activity.

**Institutional Review Board Statement:** The study was conducted according to the guidelines of the Declaration of Helsinki, and approved by the Ethics Committee of Regional Hospital of Bolzano (n° 94-2017) and by the University of Trento Human Subjects Research Committee (n° 2017-031).

**Informed Consent Statement:** Informed consent was obtained from all subjects involved in the study.

**Data Availability Statement:** The data presented in this study are available on request from the corresponding author. The data are not publicly available due to privacy information.

**Acknowledgments:** We are grateful to the Fondazione Trentina per la Ricerca sui Tumori for funding this project, within the main project on cancer pain. We thank the Rheuma-Liga Südtirol, which, after this project, with contribute of Mut Social Foundation, promoted the interdisciplinary Outpatient Clinic for Developmental Age Rheumatology. Lastly, we wish to thank all the patients, family members, and staff from all the units that participated in the study.

**Conflicts of Interest:** The authors declare no conflict of interest. The funders had no role in the design of the study; in the collection, analyses, or interpretation of data; in the writing of the manuscript, or in the decision to publish the results.

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
