# Peer review of "SMART4Pain: Feasibility of a Two-Arm Pilot Study of an Integrated Rehabilitation Program for Adolescents and Their Parents for Improving Pain Management"

_adolescents, doi:10.3390/adolescents1040037_

Round 1

Reviewer 1 Report

The study design includes Group 1 and Group 2, i.e., CBT vs. biofeedback. To draw the conclusions, the authors need to have three-arm comparisons. Add additional group with  Rational Emotive Education only.

The number is small to generate and support the final analysis. The authors should recruit more samples.

Reviewer 2 Report

After reading this manuscript, I have two main concerns:

1) as noted above, I have concern about the English language used, including use of punctuation, as it does not flow well and makes the manuscript difficult to understand. General issues include too many commas within sentences, word choices that are not appropriate or are unnecessary (such as too many descriptors when fewer would be sufficient), and words being translated incorrectly (such as informations instead of information, adhesion versus adherence, young individual versus youth)

2) the study described provides very little additional content to the literature. Specifically, it is well established that CBT and biofeedback are helpful for pain management, so it is very unclear why the authors chose to divide the participants into two groups for the intervention. The online modules could be valuable to the literature, however, the description of the modules within the manuscript was minimal. I would recommend reworking the manuscript to focus more on the online modules than the different groups (CBT and biofeedback); including how these online modules are different from others offered in different countries. 

Related more minor comments:

1) the first two sentences in the introduction are contradictory to each other; either the effects of pain are uncertain or the pain causes impaired functioning in various areas of life

2) page 3 line 110 - exclusion criteria about not having access to a web-based device was already accounted for in the inclusion criteria

3) page 4 lines 156-158 - this description of biofeedback is unnecessary given the description in lines 159-160

4) page 5 line 180 - completion of study assessments is not assessing the feasibility of the program, those assessments are only required for the study

5) page 6 - lines 215 and 217- use the word analgesic instead of pain killer (this is an example of inappropriate translation to English, I believe)

6) page 6 lines 232 -233 - why include the breakdown of the response scale for only this measure, while the other measures only have the range described

7) page 7 line 241 - why is only the self-report scale included for analysis? 

8) page 8 lines 304 - I’m unclear about participation rate being 74%…is this from the original 27 possible participants? If so, then the description of the sample should say 27 adolescents, not 20. 

9) the lack of follow up after the 6-8 weeks is a HUGE flag that this study was not conducted in a very thorough manner. There is a comment about this in the discussion, but it is not clear what happened to interfere with follow up occurring or even if follow up was originally planned and could not be completed

10) please include an explanation about why there are no specific outcomes for the CBT group as there are for the biofeedback group

11) at various points in the manuscript there is mention for the program being “tailored” for the participants, however, on page 12 line 385 the statement indicates that “all participants” received the same content. Therefore, it is unclear if the program was individualized or standardized

12) page 2 line 83 - a citation should be provided fro the interventions working well in other countries; which goes back to my main concern about this study not adding much to the literature at this time 

Round 2

Reviewer 2 Report

It is clear that the authors have taken into consideration most of the comments in the previous review. The current manuscript is now easier to read and the presented results are more clearly connected to the introduction. However, I still have concerns about the overall merit of the manuscript. The study hypothesis focuses on the the combination of internet and in-person treatment overall, so it is still unclear what is added to the literature by having the participants divided into two different in-person groups. 

There are still a few sentences (such as page 2 line 50) that are slightly confusing to read in English. 
